# Circular RNA circ_0001591 Contributes to Melanoma Cell Migration Through AXL and FRA1 Proteins by Targeting miR-20a-3p and miR-34a-5p

**DOI:** 10.3390/genes16080921

**Published:** 2025-07-30

**Authors:** Elisa Orlandi, Elisa De Tomi, Francesca Belpinati, Marta Menegazzi, Macarena Gomez-Lira, Maria Grazia Romanelli, Elisabetta Trabetti

**Affiliations:** 1Section of Biology and Genetics, Department of Neurosciences, Biomedicine and Movement Sciences, University of Verona, Strada Le Grazie, 8, 37134 Verona, Italy; elisa.orlandi@univr.it (E.O.); elisa.detomi@univr.it (E.D.T.); francesca.belpinati@univr.it (F.B.); mariagrazia.romanelli@univr.it (M.G.R.); elisabetta.trabetti@univr.it (E.T.); 2Section of Biological Chemistry, Department of Neurosciences, Biomedicine and Movement Sciences, University of Verona, Strada Le Grazie, 8, 37134 Verona, Italy; marta.menegazzi@univr.it

**Keywords:** circular RNAs, MicroRNAs, melanoma, miR-20a-3p, miR-34a-5p, Axl, Fra1, circ_0001591

## Abstract

**Background/Objectives:** Different risk factors are involved in the initiation and progression of melanoma. In particular, genetic and epigenetic pathways are involved in all stages of melanoma and are exploited in therapeutic approaches. This study investigated the role of circular RNA circ_0001591 in melanoma cell migration. **Methods:** Three different melanoma cell lines were transfected with siRNA targeting circ_0001591 and with mimic or inhibitor molecules for miR-20a-3p and miR-34a-5p. Gene and protein expression levels were analyzed by RT-qPCR and Western blot, respectively. Dual luciferase reporter assays were performed to confirm the direct interaction of miR-20a-3p and miR-34a-5p with circ_0001591, as well as with the 3’UTRs of AXL (for both miRNAs) and *FOSL1* (miR-34a-5p only). Wound healing assays were conducted to assess cell migration velocity. **Results:** The silencing of circ_0001591 significantly reduces the migration ability of melanoma cell lines. This downregulation was associated with an increased expression of miR-20a-3p and miR-34a-5p. Dual luciferase reporter assays confirmed the direct binding of both miRNAs to circ_0001591, supporting its role as a molecular sponge. The same assays also verified that miR-20a-3p directly targets the 3’UTR of *AXL*, while miR-34a-5p binds the 3’UTRs of both *AXL* and *FOSL1*. Western blot analysis showed that the modulation of this axis affects the expression levels of the AXL and FRA1 oncoproteins. **Conclusions:** Our findings demonstrate that circ_0001591 promotes melanoma migration by sponging miR-20a-3p and miR-34a-5p, thereby indirectly modulating the expression of AXL and FRA1 oncoprotein. Further investigations of this new regulatory network are needed to better understand its role in melanoma progression and to support the development of targeted therapies.

## 1. Introduction

Melanoma is the most aggressive type of skin cancer. Although it accounts for only 1% of all cases, it is responsible for more than 80% of deaths caused by cutaneous malignancies [1]. It represents 1.7% of cancers identified and ranks as the fifth most common cancer diagnosed [2]. In recent years, the molecular characterization of melanoma has become an important factor for helping doctors towards personalized therapeutic treatments. Tumor cells are characterized by heterogeneous genetic and epigenetic profiles; as a result, all these alterations can affect the effectiveness of traditional treatments, with a direct impact on recurrence and prognosis [3,4]. In melanoma, primary targets such as BRAF, MEK, NRAS, and c-KIT are currently used for targeted therapies [5]. However, despite the different molecular pathways and the development of various therapeutic approaches against these targets, melanoma remains a complex disease [3,5]. For this reason, it is essential to extend the spectrum of molecular analysis to include other emerging targets, such as AXL and FRA1, which are distinguished by the fact that they do not depend on genetic mutations but on overexpression mechanisms.

AXL is a transmembrane receptor belonging to the TAM tyrosine kinase receptor family. It has been reported to be one of the TAM receptors overexpressed in many human tumors, including melanoma [6]. AXL signaling is involved in the regulation of inflammatory response and plays a role in the immune activation of smooth muscle cells [6]. Once activated, AXL promotes the growth and survival of numerous cell types through different pathways, including MAPK/ERK and PI3K/AKT [7]. Several studies have identified AXL as a potential target for new therapies, as its inhibition leads to verified anti-tumor activity in vivo in several types of cancers, including melanoma (Figure 1) [8].

The FOS-related antigen 1 (FRA1) protein is a transcription factor encoded by the FOS-like 1 gene (*FOSL1*). It is a member of the activator protein-1 (AP-1) transcription factor superfamily and the FOS family of proteins [9]. *FRA1*, like *AXL*, has been found to be an overexpressed gene in melanoma [10]. The high expression of *FRA1* in melanoma has been confirmed to be capable of inducing both the activation of the RAS-BRAF-MEK-ERK pathway and the expression of transcription factors related to epithelial–mesenchymal transition (EMT). This ability causes epithelial cells to acquire mesenchymal, invasive, and tumorigenic capabilities. In this way, it drives the reprogramming of tumor cells, leading to the dedifferentiation of melanocytes (Figure 1) [11].

Since the overexpression of genes such as *AXL* and *FOSL1* plays an important role in resistance to traditional therapies in melanoma and other types of tumors, part of the scientific research has focused on the regulatory mechanisms activated upstream by non-coding RNAs. Among these, circular RNAs (circRNAs) have been shown to sequester microRNAs, key repressors of gene expression, thereby attenuating their target genes from inhibition. As a result, the accumulation of circRNAs can amplify oncogenic signaling pathways [12,13,14]. Several studies suggest that circRNAs are involved in invasion, apoptosis, and resistance to therapy [15], and many of them are considered oncogenic [16,17,18]. In melanoma, they are associated with prognosis and disease progression stages [16,19], and evidence also suggests that specific circRNAs have been implicated in the alteration of sensitivity to target therapy or immunotherapy [20,21]. Among those identified in melanoma, serum circ_0001591 is associated with both survival and disease-free survival in melanoma patients. The authors demonstrated that circ_0001591 promotes cell growth and invasion and reduces the apoptotic rate in melanoma cell lines by sponging miR-431-5p, causing the upregulation of the ROCK1/PI3K/AKT (Rho Associated Coiled-Coil Containing Protein Kinase 1/Phosphoinositide 3-kinase/Protein kinase B) signaling pathway [22].

To further investigate the functional role of circ_0001591 in melanoma, from the numerous predicted miRNA targets, we focused our attention on two microRNAs: miR-20a-3p and miR-34a-5p.

The first, miR-20a-3p, is a member of the miR-17-92 cluster, a family of fifteen microRNAs [23]. Although many studies have focused on this cluster, miR-20a-3p remains the less explored one. Evidence suggests that miR-20a-3p acts as a neuroprotective factor in stroke and stroke-related outcomes [24,25]. In cancers, miR-20a-3p plays a dual role, depending on the context. For example, in colon–rectal cancer, it is associated with metastasis [26], whereas in oral squamous cell carcinoma [27] and pancreatic ductal adenocarcinoma, it appears to act as a tumor suppressor. Moreover, low expression levels of this miRNA have been associated with worse prognosis in glioblastoma [28]. The second, miR-34a-5p, is a well-characterized microRNA that belongs to the miR-34 family, which is known to play a tumor suppressor role in various cancer types. When upregulated, members of this family can induce apoptosis, cell-cycle arrest, and senescence [29]. Additionally, they can negatively influence cancer stem cell viability and metastasis formation [29]. Many studies have demonstrated the role of miR-34a-5p in cancer [30,31,32], and circRNAs have been shown to regulate its expression [33,34,35].

This study showed that circ_0001591 can participate in melanoma migration through the sponging of miR-20a-3p and miR-34a-5p, which, in turn, results in the upregulation of oncoproteins AXL and FRA1.

## 2. Results

### 2.1. Circ_0001591 in Melanoma Cell Migration

Initially, to ensure the amplification of circ_0001591 by qRT-PCR, the A375 RNA was treated with poly-A polymerase and RNase R. Following this treatment, cDNA synthesis was carried out, and qRT-PCR (quantitative real-time PCR) was performed using divergent primers for circ_0001591 and convergent primers for its parental gene, *H2AC11* (H2A Clustered Histone 11). As expected, the linear *H2AC11* gene was digested, and the expression of circ_0001591 was unmodified by the digestion with RNase R, demonstrating its circularity and stability (Figure 2A).

The A375, LM-36, and LM-20 melanoma cell lines were transfected with si-circ_0001591, and qRT-PCR was performed at different times after transfection. CircRNA significant inhibition was achieved in all three cell lines after 8–16 h (Figure 2B–D).

To verify whether the impact of circ_0001591 inhibition had a direct effect on tumor migration velocity, a wound healing assay was performed. In the A375 and LM-36 melanoma cell lines, a significant reduction in the migration rate was observed 24 h after siRNA (silencing RNA) transfection (Figure 2E,F). In the LM-20 cell line, circRNA inhibition showed a significant decrease in melanoma migration at all time points (Figure 2G).

### 2.2. Circ_0001591 Targeted miR-20a-3p and miR-34a-5p

The CircMine web application was used to predict miRNAs that could interact with circ_0001591. Its circRNA-miRNA prediction tool showed that cric_0001591 contains two putative binding sites for miR-20a-3p and two for miR-34a-5p. To experimentally validate these results, we tested whether the inhibition of circ_0001591 could affect the expression levels of these miRNAs. The results of the transfection of circ-specific siRNA showed an increase in miR-20a-3p and miR-34a-5p levels 8–24 h after treatment, supporting the hypothesis that circ_0001591 may function as a sponge for both miRNAs. In detail, A375 cells showed the significant overexpression of miR-20a-3p at 8 and 12 h and miR-34a-5p from 8 to 24 h after transfection. In LM-20 cells, the significant miRNAs upregulation could be observed at 8–16 h for miR-20a-3p and 8–12 h for miR-34a-5p. Finally, in the LM-36 cell line, the inhibition of circ_0001591 resulted in a significant increase in miR-20a-3p levels at 16 and 24 h and miR-34a-5p expression at 12 and 16 h after transfection (Figure 3A–C).

Dual luciferase reporter assay determined the direct interaction of circ_0001591 with miR-20a-3p and 34a-5p. Four plasmids were constructed, one for each sponge region of the circ_0001591; the schematic representation of the wild-type and mutated plasmids is shown in Figure 4A–D.

Compared to the wild-type plasmids, the ones containing the mutated seed regions demonstrated a higher luciferase activity, indicating that the selected sequences might be targets of a direct interaction between the circRNA and the two miRs (Figure 4E–G).

### 2.3. miR-20a-3p and miR-34a-5p in Melanoma Cell Migration

To explore whether miR-34a-5p and/or miR-20a-3p could affect cell mobility, the miRNA mimics or their inhibitors were transfected into the three cell lines, and a wound healing assay was performed. The transfection efficiency was evaluated through qRT-PCR analysis (Figure 5A,B).

The results obtained in the wound healing assay show that overexpression of the microRNAs in the study reduced melanoma migration in A375, LM-20, and LM-36 cell lines (Figure 6A–C), while their inhibition reversed this effect (Figure 6D–F). These findings indicate that both microRNAs are involved in the melanoma cells’ mobility and could participate in invasion and metastasis.

### 2.4. miR-20a-3p and miR-34a-5p Target AXL and FOSL1 Genes and Negatively Regulate Axl and Fra1 Proteins

To further investigate the functional role of these miRNAs, the web-based application mirDB was used to investigate the potential mRNA target downstream of the circRNA-microRNA interaction. The analysis revealed one potential seed region for miR-34a-5p within the 3’UTR of the *AXL* gene and two seed regions within the 3’UTR of the *FOSL1* mRNA. Although no binding sites were shown in mirDB between miR-20a-3p and *AXL*, manual sequence analysis of *AXL* 3’UTR revealed a complementary sequence corresponding to the potential binding site of miR-20a-3p, whereas no seed region for miR-20a-3p was found in the 3’UTR of FOSL1.

To explore whether miR-34a-5p and miR-20a-3p could regulate AXL and FRA1 expression, melanoma cell lines were transfected with miRs mimics or inhibitors. After a 24 h incubation period, cells were analyzed by qRT-PCR to determine whether the miRs affected the transcription or stabilization of *AXL* and *FOSL1* mRNA and by Western blot to investigate their effects upon protein translation. Moreover, to validate the direct interaction between *AXL* and *FOSL1* 3’UTRs and the two studied miRNAs, a dual luciferase reporter assay was performed.

Expression analyses of mRNAs showed a statistically significant impact of both microRNAs on *AXL* transcription or stability only in the LM-20 cell line, while no other significant effects were observed in the other cell lines (Figure 7A–C).

Although significant modulation of mRNA levels was observed only in LM-20, Western blot analyses showed that the transfection of miRNA mimics resulted in a significant reduction in AXL and FRA1 protein expression across all three melanoma cell lines. To confirm the regulatory effect of these microRNAs, the transfection of specific inhibitors reversed this effect, leading to an increase in the protein levels of these oncoproteins (Figure 8A–C).

Figure 9 shows the schematic representation of the miR-20a-3p and miR-34a-5p binding sites in *AXL* and *FOSL1* mRNAs (Figure 9A–D).

To validate the direct interaction between *AXL* 3’UTR and miR-20a-3p and miR-34a-5p, and between *FOSL1* 3’UTR and miR-34a-5p, a dual luciferase report assay was performed, confirming that mutagenized plasmids exhibited higher luciferase activity, compared to the wild-type plasmids (Figure 9E–G). In particular, in *FOSL1* 3’UTR, the mutagenesis of the second binding site of miR-34a-5p results in more luciferase activity than the first one. But in A375 and LM-36, the mutagenesis of both binding sites results in lower luciferase activity than the second binding site.

## 3. Discussion

Diagnosis of early-stage melanoma makes the tumor surgically resectable without further treatment; thereafter, melanoma is associated with a high probability of developing metastasis and resistance to standard therapies [36]. Despite the advent of novel treatment in the last few years, metastatic melanoma remains a malignancy with a poor prognosis [37]. Somatic mutations in genes encoding proteins, such as MAPK, p53, and PI3K, have been investigated, and they appear to be drivers of metastatic progression. However, recent advances have revealed a complex landscape of genomic and epigenetic abnormalities that need further examination [38,39,40]. For example, AXL and FRA1 are two emerging potential targets that are overexpressed in melanoma and are important in several signaling regulator pathways involved in melanoma progression and metastasis [10,41]. Moreover, recent studies have shown that their inhibition has synergistic effects with current therapies used for melanoma [42,43].

In current cancer research, circRNAs have emerged as an important area of investigation due to their involvement in the initiation, progression, and metastasis of different types of cancer. Their primary function is to regulate gene expression by acting as a miRNA sponge, essentially binding to and inhibiting the activity of miRNAs [15].

In melanoma, several circRNAs play an oncogenic role since their upregulation promotes cell progression and tumor growth [44]. For example, circ_0084043 was a candidate to upregulate SNAILl (Snail Family Transcriptional Repressor 1) by interacting with miR-153-3p, thereby promoting melanoma cells’ proliferation, invasion, and migration [16]. The same circRNA can sponge miR-429, and its knockdown inhibited the WNT/β-CATENIN signaling pathway via the miR-429/TRIB2 (tribbles homolog 2) axis [45]. Another example is circ_0079593, which is involved in the development and aggressiveness of melanoma by regulating *CHAF1B* (Chromatin Assembly Factor 1 Subunit B) and *MCAM* (Melanoma Cell Adhesion Molecule) genes through the inhibition of miR-516b-5p [46].

Inspired by this evidence, we analyzed the role of circ_0001591, a circRNA recently shown to be upregulated in melanoma and negatively associated with overall survival. Yin and colleagues demonstrated that this circ_0001591 sponges miR-431-5p, resulting in the upregulation of the ROCK1/PI3K/AKT pathway, and this is associated with cell growth and invasion in melanoma cell lines [22].

In our study, we investigated that circ_0001591 might act as an oncogene in three cell lines with different grades of malignancies: A375, which is a cutaneous melanoma cell line; LM-20, a nodal metastasis cell line; and LM-36, a cutaneous metastasis cell line. We hypothesize that this may occur through the sponge of miR-20a-3p and miR-34a-5p, resulting in the upregulation of the oncoproteins AXL and FRA1.

Regarding miR-20a-3p, evidence indicates that it functions as a tumor suppressor in oral squamous cell carcinoma because it reduces the migration and proliferation of a cell model, inhibiting STAT3 (Signal Transducer And Activator Of Transcription 3) expression [47]. Moreover, miR-20a-3p showed an anti-proliferative effect in the melanoma cell line B16; its upregulation decreased cellular growth [48]. As for miR-34a-5p, it has been shown to inhibit the growth and migration in uveal melanoma cell lines by targeting c-Met (MET Proto-Oncogene, Receptor Tyrosine Kinase) [49]. The same tumor suppressive role of miR-34a-5p has been demonstrated in several different types of cancer, including head and neck squamous cell carcinoma [50], triple-negative breast cancer [51], and osteosarcoma [52].

Since cell migration plays a crucial role during the whole cascade of cancer progression and is particularly significant during invasion, the initial step of metastasis [53], we performed a wound healing assay after the knockdown of circ_0001591 by siRNA transfection. The results revealed a reduction in the migration of A375, LM-20, and LM-36 melanoma cell lines, suggesting that circ_0001591 may contribute to melanoma invasion and metastasis. Moreover, the silencing of circ_0001591 resulted in the upregulation of miR-34a-5p and miR-20a-3p, with a significant increase across all three melanoma cell lines, showing that circ_0001591 can sponge these miRNAs. Previous studies have reported similar miRNA-circRNA interactions, demonstrating that miR-20a-3p is sponged by circ_0005105 in pancreatic ductal adenocarcinoma [53], and miR-34a-5p is sponged by circ_0036602 in cervical cancer [34] and by circ_ITGA7 in glioma [54]. We demonstrated direct interaction between circRNA and both miRs by the higher expression of luciferase activity following the transfection of vectors carrying the mutated sponge binding regions in circ_0001591.

Moreover, we show that the overexpression of miR-20a-3p and miR-34a-5p led to a significant decrease in cell migration, and transfection with their inhibitors increased migratory capacity, indicating that these miRNAs may play a key role in melanoma invasion and metastasis by affecting cell motility.

As to the close relation of the AXL and FRA1 proteins with the invasion, migration, and proliferation in melanoma cell lines [10,55], we investigate whether these genes could be targeted by miR-20a-3p and/or miR-34a-5p. In silico analysis showed a binding site for miR-34a-5p within the *AXL* 3’UTR sequence. Two binding sites for miR-34a-5p were identified within the *FOSL1* 3’UTR, while no binding site for miR-20a-3p was found within the *AXL* and *FOSL1* 3’UTR.

qRT-PCR showed that in LM-20, the overexpression of miR-34a-5p or miR-20a-3p affected *AXL* transcription or promoted mRNA decay, while no effect was seen in LM-36 or A375 cells. Instead, Western blot analysis demonstrated that the expression of AXL and FRA1 was reduced by the overexpression of both miRs, showing their impact on protein translation. Surprisingly, the level of the FRA1 protein was also affected by miR-20a-3p transfection. This result indicates that a different pathway regulated by miR-20a-3p can affect FRA1 expression since *FOSL1* 3’UTR presents no binding site for this miR.

*AXL* mRNA is a direct target of miR-34a-5p in lung cancer, showing that the overexpression of miR-34a-5p suppresses *AXL* expression and inhibits cell migration and invasion [56] in triple-negative breast cancer, leading to reduced proliferation and invasion [57], and other cancers, including melanoma [58]. In the present study, we demonstrated that, as well as in other cancers, *AXL* is a direct target of miR-34a-5p in the three melanoma cell lines studied. Moreover, we demonstrate, for the first time, the direct regulation of *AXL* mRNA by miR-20a-3p, due to the presence in *AXL* 3’UTR of a sequence corresponding to the miR-20a-3p seed region. Concerning the *FOSL1* gene, our results show that in three melanoma cells, miR-34a-5p directly regulates its expression by binding to the 3’UTR region. Specifically, we describe two binding sites for miR-34a-5p in *FOSL1,* and we found that mutation of the first binding site resulted in a modest but significant increase in luciferase activity. In contrast, mutation of the second site led to a more pronounced increase, suggesting that this site plays a more critical role in regulating protein translation. Interestingly, mutating both binding sites did not lead to a further increase in luciferase expression.

## 4. Materials and Methods

### 4.1. Cell Cultures

A375 cells (CRL-1619) were purchased from ATCC (Manassas, VA, USA). LM-20 (17697M) and LM-36 cells (26414M) were provided by Dr. Monica Rodolfo (Istituto Nazionale Tumori Milano (MI), Italy) and derived from nodal and cutaneous metastasis biopsies of melanoma, respectively [59]. All cell lines used carry the V600E mutation in the *BRAF* gene. When not specified, all the cell lines were cultured in Roswell Park Memorial Institute 1640 medium (RPMI), supplemented with 10% heat-inactivated fetal bovine serum (FBS), 1% L-glutamine (200 nM solution), and 2% penicillin–streptomycin (5000 IU/mL and 5000 µg/mL solutions, respectively) (Gibco, Grand Island, New York, NY, USA). Cell cultures were incubated at 37 °C in a humidified atmosphere containing 5% CO_2_. A375, LM-20, and LM-36 cell lines were tested for *Mycoplasma* contamination.

### 4.2. siRNAs, Mimics, and Inhibitors Transfections

Two hundred thousand cells from each cell line were seeded per well in 12-well plates, and after 24 h, transfections were carried out. SiRNA, mimics, inhibitors, and negative control transfections were performed using 0.5 µL of Lipofectamine 3000, following the manufacturer’s instructions (Thermo Fisher Scientific, Waltham, MA, USA). For siRNA transfections, 50 nM of two siRNAs, designed to interact specifically with the back splice junction, were used. The siRNA and negative control sequences used are listed in Table 1 (GenePharma, Shanghai, China). Cells were collected 4, 8, 12, 16, and 24 h after transfection for RNA analysis. Three independent experiments were carried out for each transfection.

The miR-20a-3p and miR-34a-5p mimics or the mimic negative control were transfected at a concentration of 50 nM; miR-20a-3p and miR-34a-5p inhibitors or the inhibitor negative control were transfected at a concentration of 100 nM (CliniSciences, Nanterre, France). Following transfections, cells were harvested 24 h later for RNA and protein extractions.

### 4.3. RNA Extraction and RNase R Treatment

Total RNA was extracted from transfected cells using Trizol, in accordance with the manufacturer’s instructions (Invitrogen, Waltham, MA, USA). A poly-A polymerase (ACRO BioSystem, Newark, DE, USA) was used to add a long tail of adenine nucleotides at the 3′ end of messenger RNA using ATP as a donor. This passage is essential to allow the digestion of *H2AC11*, the circ_0001519 parental gene, by RNase R, due to its highly structured RNA composition. Indeed, it is well-established that RNase R cannot digest RNAs with highly structured ends [60]. The poly-A polymerase treatment was performed for 1 h at 37 °C, with 5 U of enzyme and 5 µg of RNA. Then, 3.3 µg of the treated RNA was digested with 10 U of RNase R (abm, Richmond, BC, Canada) for 45 min at 37 °C.

### 4.4. cDNA Synthesis and Quantitative Real-Time PCR

The SensiFAST cDNA Synthesis Kit (Bioline, London, UK) was used for the first-strand cDNA synthesis of mRNAs; 500 ng of each RNA sample was retrotranscribed following the manufacturer’s instructions. cDNA quantification was performed using quantitative real-time PCR (qRT-PCR) on the CFX Connect Real-Time System (Bio-Rad, Hercules, CA, USA). The SensiFAST SYBR No-ROX kit (Bioline, London, UK) was used to quantify the *AXL*, *FOSL1*, circ_0001591, and *H2AC11* gene expression. The TATA BOX binding protein (*TBP*) gene was employed as a normalizer to calculate the relative quantification using the Delta–Delta Ct method (2^−ΔΔCt^) [61]. The primers used are listed in Table 2; divergent primers were devised to amplify circ_0001591 and were specifically designed to amplify the back-splice junction sequence, thereby avoiding the amplification of the linear mRNA from which the circRNA is derived.

The TaqMan Advanced miRNA cDNA Synthesis Kit (Thermo Fisher Scientific, Waltham, MA, USA) was used to reverse transcribe microRNAs. One µL of each RNA sample was utilized to retrotranscribe miRNAs according to the manufacturer’s instructions. Specific TaqMan Advanced miRNA Assays (Thermo Fisher Scientific, Waltham, MA, USA) were employed to quantify miR-20a-3p and miR-34a-5p, using miR-191-5p as a normalizer. Relative quantification was conducted using the Delta–Delta Ct method (2^−ΔΔCt^). The specific probes used are listed in Table 2.

### 4.5. Protein Extraction and Western Blot Analysis

For protein extraction, cells from each experiment were lysed using RIPA buffer and protease inhibitors (Mirus Bio LLC, Madison, WI, USA), and the concentration was determined by Bradford assay. In total, 20 µg of total protein per sample were subjected to SDS polyacrylamide gel electrophoresis (SDS-PAGE) using a 10% gel and electroblotted onto PVDF membranes (Thermo Fisher Scientific, Waltham, MA, USA). Then, membranes were blocked in 5% non-fat milk in TBST solution. Membranes were hybridized overnight at 4 °C with primary antibodies anti-AXL (1:3000), anti-FRA1 (1:1000), or anti-TBP (1:2000) (ABclonal, Woburn, MA, USA). The next day, membranes were incubated with anti-rabbit HRP-conjugated secondary antibody (Cell Signaling Technology, Danvers, MA, USA). Bound antibody signal detection was performed using chemiluminescence with a WesternBright ECL kit (Advansta, San Jose, CA, USA). The resulting images were acquired with an Azure C300 Processing machine (Azure Biosystem, Dublin, CA, USA). ImageJ 1.53 software was used for densitometric analysis, and the protein quantification was performed using TBP as a normalizer.

### 4.6. Wound Healing Assay

A 12-well plate was seeded with 200,000 cells per well for each melanoma cell line. Once the cells reached confluence, the monolayer was scratched with a sterile 10 µL pipette tip. Detached cells were removed by washing with medium, after which fresh medium containing 2% FBS was added. Immediately after the scratch, the cells were transfected as described in “siRNAs, mimics, and inhibitors transfections” and then incubated at 37 °C in a humidified atmosphere with 5% CO_2_. At 0, 24, 48, and 72 h, an image of each well was captured using an inverted microscope (Axio Vert A1, Zeiss, Oberkochen, Germany). The wound healing assay was performed starting from 24 h, as in general, 24–96 h is the ideal timeframe for investigating the functional effects of the siRNA knockdown in cell culture [46]. All images were then analyzed using ImageJ software to measure the scratched area.

### 4.7. Plasmid Construction and Mutagenesis

Four different fragments of circ_0001591 were amplified using primers that target the two binding sites for miR-34a-5p and the two binding sites for miR-20a-3p. Two other fragments were amplified, corresponding to regions of the *AXL* 3’UTR containing the miR-20a-3p and miR-34a-5p binding sites. The entire 3’UTR of the *FOSL1* gene, which contains two binding sites for miR-34a-5p, was also amplified. The primers utilized are listed in Table 3. The fragments amplified were introduced into the pGL3 promoter vector plasmid. The correct sequences of all plasmid vectors were confirmed by Sanger sequencing (BMR Genomic, Padova, Italy).

Mutagenesis of the plasmid containing the miR-20a-3p or miR-34a-5p seed regions was performed with the QuikChange Site-Directed Mutagenesis Kit (Stratagene, Agilent Technologies, La Jolla, CA, USA), according to the manufacturer’s instructions. The primers used for the mutagenesis are listed in Table 4. The correct mutagenesis was confirmed by Sanger sequencing (BMR Genomic, Padova, Italy).

### 4.8. Dual Luciferase Reporter Assay

The plasmid constructs were transfected into cells by Lipofectamine 3000, following the manufacturer’s instructions (Thermo Fisher Scientific, Waltham, MA, USA). A total of 120,000 cells were seeded in a 24-well plate and cultured for 24 h. Wild-type or mutagenized constructs were co-transfected with either miR-20a-3p or miR-34a-5p, depending on the specific seed region of the construct used. The pRL null vector was used as a control. After a 24 h incubation period, the Firefly and Renilla luciferase activities were assessed by the dual luciferase assay reporter system following the manufacturer’s instructions (Promega, Milan, Italy), and the ratio of Firefly/Renilla luciferase activity was calculated.

### 4.9. In Silico Analysis

The bioinformatic web-based application CircMine (http://www.biomedical-web.com/circmine/home accessed on 15 September 2022) was used to predict potential interactions between circ_0001591 and the microRNAs miR-34a-5p and miR-20a-3p. Furthermore, the miRDB database (https://mirdb.org/ accessed on 21 January 2023) was used to predict the interaction of miR-34a-5p and miR-20a-3p with the 3’UTR of *AXL* and *FOSL* genes. The circRNA-microRNA-mRNA interactions analyzed in this study were selected because they showed biological relevance according to the scientific literature.

### 4.10. Statistical Analysis

All data were entered into GraphPad Prism version 7.03, and differences between conditions in the experiments were calculated using the statistical tool and an unpaired, two-tailed Student’s *t*-test. One or two asterisks indicate a *p*-value of less than 0.05 or 0.01. The results are presented as means with standard deviations of a minimum of three separate biological replicates.

## 5. Conclusions

In conclusion, our results demonstrate that circ_0001591 plays a role in melanoma cell migration. The underlying mechanism may be in part attributed to the sponge effect on miR-20a-3p and miR-34a-5p, which cannot downregulate the expression of AXL and FRA1 proteins.

Future studies using in vivo models or patient-derived samples may further confirm the involvement of circular RNAs and miRs in melanoma progression. The increasing depth of knowledge of the molecular mechanisms involved could lead to the definition of new therapies for developing personalized medicine.

## Figures and Tables

**Figure 1 genes-16-00921-f001:**
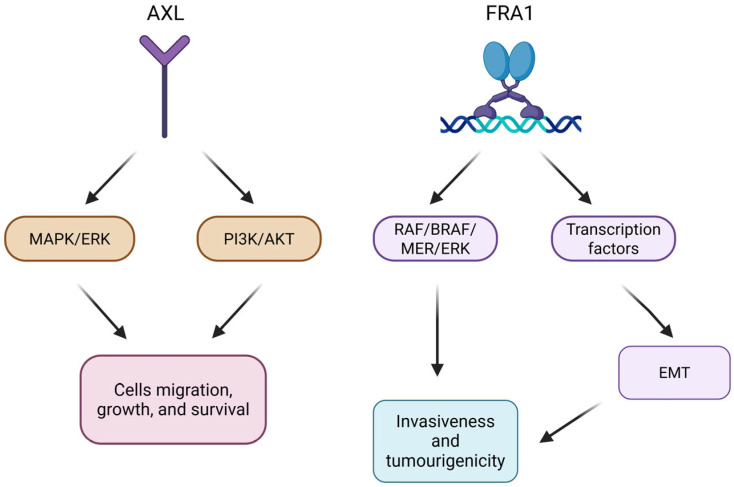
Schematic representation of AXL and FRA1 signal pathways in melanoma.

**Figure 2 genes-16-00921-f002:**
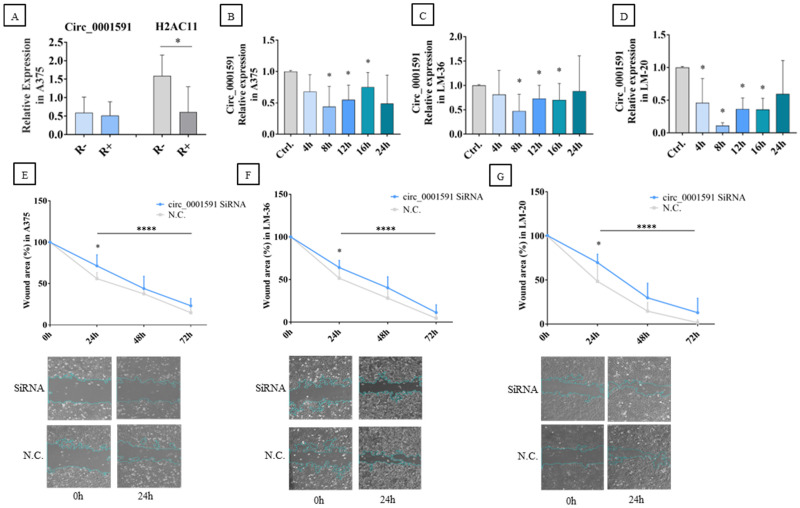
(**A**) Circ_0001591 and H2AC11 expression before and after RNase R digestion. Circ_0001591 expression after siRNA transfection at different times in (**B**) A375, (**C**) LM-36, and (**D**) LM-20 melanoma cell lines. Wound healing assay after circ_0001591 inhibition in (**E**) A375, (**F**) LM-36, and (**G**) LM-20 cell lines. SiRNA = transfection with silencing RNA; N.C. = negative control. * = *p* value < 0.05; **** = *p* value < 0.0001.

**Figure 3 genes-16-00921-f003:**
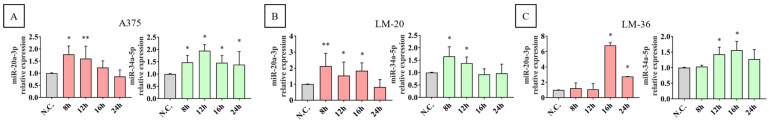
MiR-20-3p (red) and miR-34a-5p (green) expression after si-circ_0001591 transfection in (**A**) A375, (**B**) LM-20, and (**C**) LM-36 melanoma cell lines. * = *p* value < 0.05; ** = *p* value < 0.01.

**Figure 4 genes-16-00921-f004:**
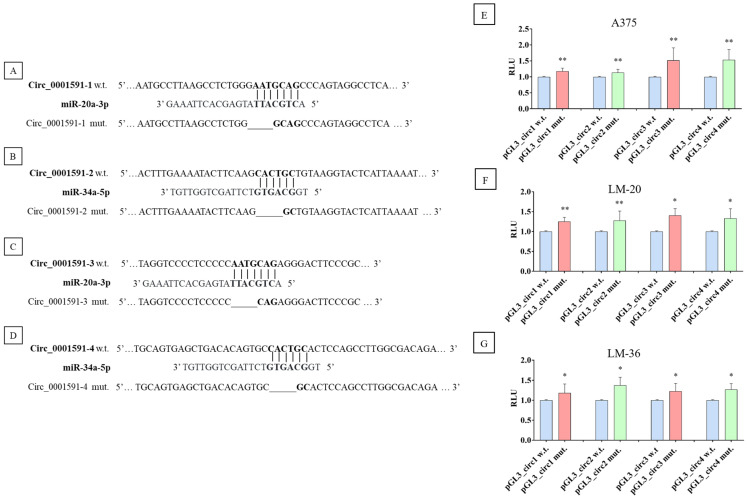
Schematic representation of miRs seed region in the wild-type plasmid (bold) and the sequence obtained by the mutagenesis of circ_0001591. The seed region wild-type and mutagenized are shown in bold, and the line shows the deletions. (**A**) The first seed region of miR–20a–3p, (**B**) the first seed region of miR-34a-5p, (**C**) the second seed region of miR–20a–3p, and (**D**) the second seed region of miR-34a-5p. Dual luciferase reporter assay results in (**E**) A375, (**F**) LM-20, and (**G**) LM-36 cell lines, after the transfection of wild-type (blue) or mutagenized plasmid for miR-20a-3p (red) or for miR-34a-5p (green). * = *p* value < 0.05; ** = *p* value < 0.01.

**Figure 5 genes-16-00921-f005:**
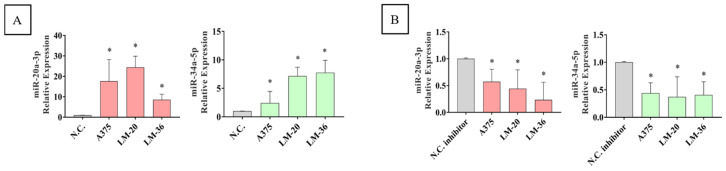
MiR-20-3p (red) and miR-34a-5p (green) expression after (**A**) mimic transfection and (**B**) inhibitor transfection, in A375, LM-20, and LM-36 cell lines, showing the transfection efficiency in all three melanoma cell lines. * = *p* value < 0.05.

**Figure 6 genes-16-00921-f006:**
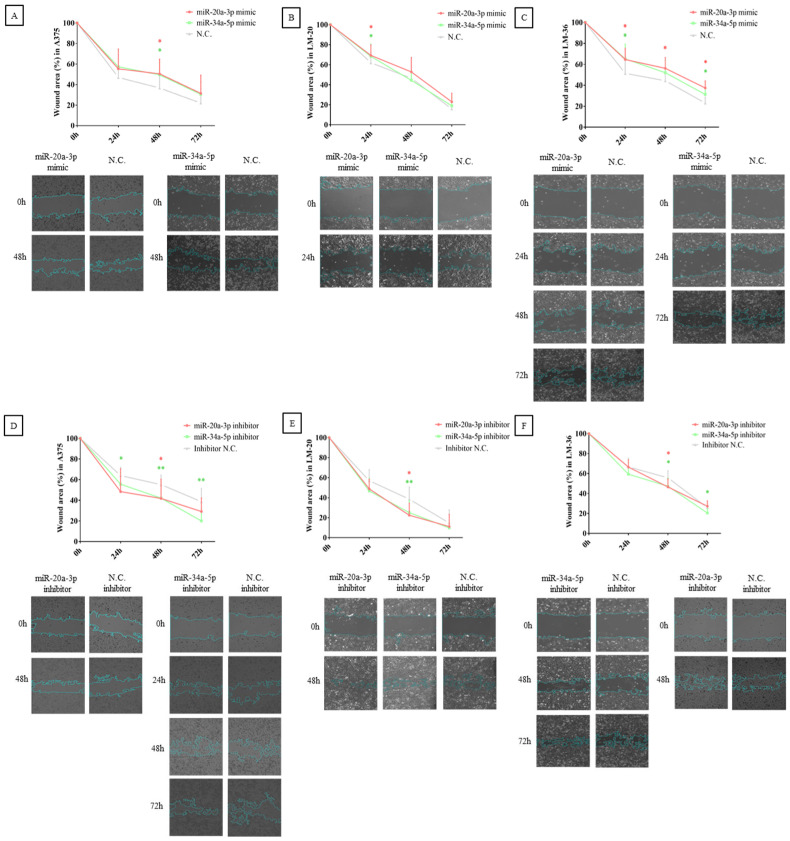
Wound healing assay after miR-20a-3p (red) or miR-34a-5p (green) overexpression in (**A**) A375, (**B**) LM-20, and (**C**) LM-36 cell lines. Wound healing assay after miR-20a-3p (red) or miR-34a-5p (green) inhibition in (**D**) A375, (**E**) LM-20, and (**F**) LM-36 cell lines. N.C. = negative control. * = *p* value < 0.05; ** = *p* value < 0.01.

**Figure 7 genes-16-00921-f007:**
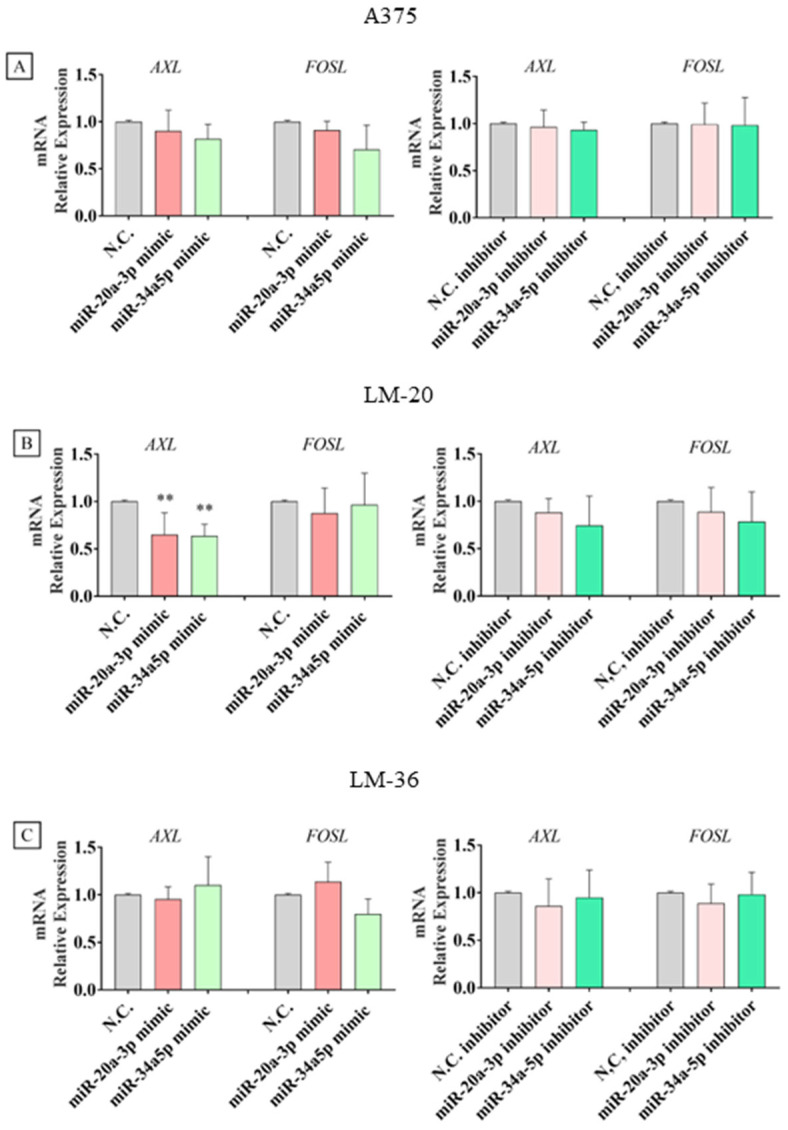
*AXL* and *FOSL1* mRNA expression after miR-20a-3p (red) or miR-34a-5p (light green) mimic transfection and after miR-20a-3p (pink) or miR-34a-5p (dark green) inhibitor transfection in (**A**) A375, (**B**) LM-20, and (**C**) LM-36 melanoma cell lines. N.C. = negative control. ** = *p* value < 0.01.

**Figure 8 genes-16-00921-f008:**
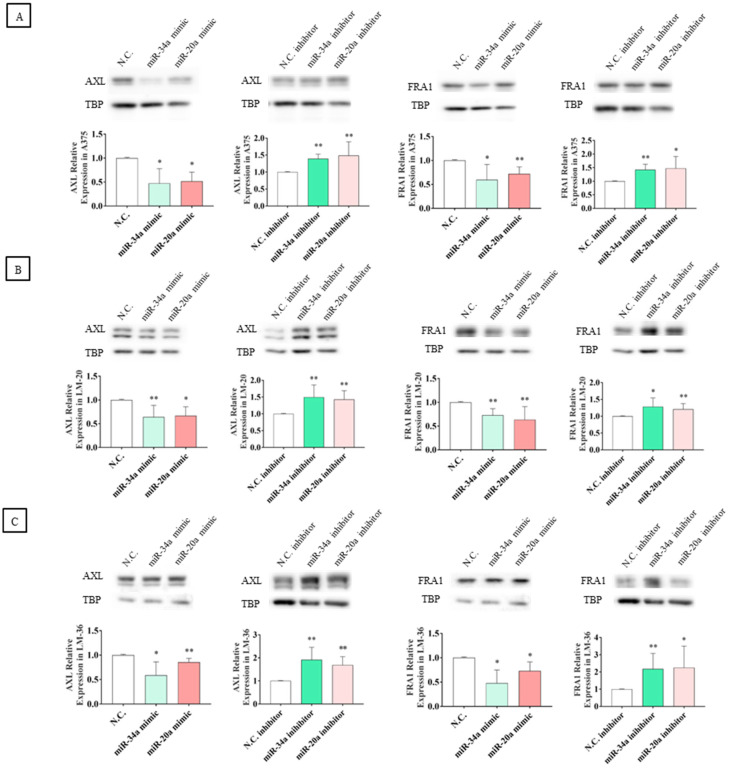
AXL and FRA1 protein expression after miR-34a-5p (light green) or miR-20a-3p (red) mimic transfection and after miR-34a-5p (dark green) or miR-20a-3p (pink) inhibitor transfection in (**A**) A375, (**B**) LM-20, and (**C**) LM-36 melanoma cell lines. N.C. = negative control. * = *p* value < 0.05; ** = *p* value < 0.01.

**Figure 9 genes-16-00921-f009:**
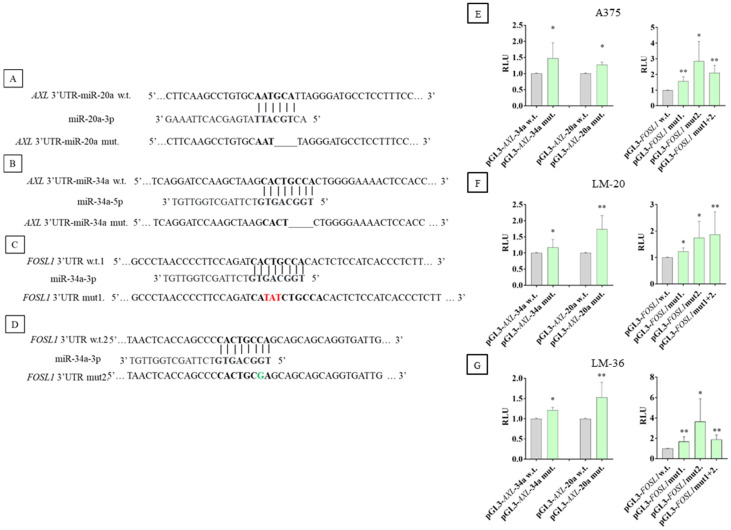
Schematic representation of miRs seed regions in the wild-type plasmid and the mutated sequence in *AXL* and *FOSL1* 3’UTR. In bold can be seen the wild-type and mutagenized seed region, a line shows a deletion, red shows nucleotide changes, and green shows an insertion. (**A**) The seed region of miR–20a–3p and (**B**) of miR-34a-5p in *AXL* 3’UTR. (**C**) The first and (**D**) second seed regions of miR-34a-5p in *FOSL1* 3’UTR. Dual luciferase reporter assay results in (**E**) A375, (**F**) LM-20, and (**G**) LM-36 cell lines, after wild-type or mutagenized plasmid transfection. The gray bars show the results of wild-type plasmids, and the green bars show the results of the mutagenized plasmids. The *AXL* results are shown in the left part of the graphs, and the *FOSL1* results are in the right part. * = *p* value < 0.05; ** = *p* value < 0.01.

**Table 1 genes-16-00921-t001:** Circ_0001591 siRNA sequences.

SiRNA	Sequence F (5′-3′)	Sequence R (5′-3′)
Hsa_circ_0001591_1	AGGCGGCAAAGCCCGCGCUAATT	UUAGCGCGGGCUUUGCCGCCUTT
Hsa_circ_0001591_2	GUGGCAAGCAAGGCGGCAAAGTT	CUUUGCCGCCUUGCUUGCCACTT
Negative Control (NC)	UUCUCCGAACGUGUCACGUTT	ACGUGACACGUUCGGAGAATT

**Table 2 genes-16-00921-t002:** Primers and probe sequences used in qRT-PCR analysis.

Gene/miR	Primers/Probe	Bp/Assay
AXL	F: 5′ ATTGGCTTCGGGATGGACAG 3′	125
	R: 5′ AAGCTCCAGGGAGGTGATTC 3′	
FOSL1	F: 5′ GCCCTTGTGAACAGATCAGC 3′	137
	R: 5′ CAGTTTGTCAGTCTCCGCCT 3′	
Circ_0001591	F: 5′ TCGCCCCCAATAGTGTTTAT 3′	136
	R: 5′ GCCCCTAGAAGAGCGAGTCT 3′	
H2AC11	F: 5′ AAGCCCAAGACTCGCTCTTC 3′	82
	R: 5′ CATAGTTGCCTTTGCGGAGC 3′	
TBP	F: 5′ TGTATCCACAGTGAATCTTGG 3′	102
	R: 5′ ATGATTACCGCAGCAAACC 3′	
hsa-miR-20a-3p	ACUGCAUUAUGAGCACUUAAAG	478317_mir
hsa-miR-34a-5p	UGGCAGUGUCUUAGCUGGUUGU	478048_mir
hsa-miR-191-5p	CAACGGAAUCCCAAAAGCAGCUG	477952_mir

**Table 3 genes-16-00921-t003:** Primers used for fragment amplification.

Plasmid	Primers (5′-3′)	Bp
Circ_0001591_1	F: CCTAAGGT TCTAGA GTCATGGCACTGGTAGGAGT	304
	R: CCTAAGGT TCTAGA AGCAAACTTTGTCAGAGGCG	
Circ_0001591_2	F: CCTAAGGT TCTAGA ACAGCAAGTTACAGCCAGTCA	317
	R: ACCTTAGG TCTAGA TGAGTTGGTTAGTCAGCACAA	
Circ_0001591_3	F: CCTAAGGT TCTAGA GTCCCCTCCCCCCAGAGGGACTTC	337
	R: ACCTTAGG TCTAGA GAAGTCCCTCTGGGGGGAGGGGAC	
Circ_0001591_4	F: CCTAAGGT TCTAGA AAGTCAGGAGTTCAGGACCAG	389
	R: ACCTTAGG TCTAGAG GCAAATCTCACAACTCAATGCC	
AXL_miR-20a-3p	F: CCTAAGGT TCTAGA GTGCCCCTCTCCTTCTTAGC	215
	R: ACCTTAGG TCTAGA TATGTGACCTGAGCCCCTCT	
AXL_miR-34a-5p	F: CCTAAGGT TCTAGA AGACAACGCTCCACCTGGTA	164
	R: ACCTTAGG TCTAGA TGCTACTCCACAGAGAAGGG	
FOSL1	F: CCTAAGGT TCTAGA AACCCTCCTCGCTTTGTGAG	893
	R: ACCTTAGG TCTAGA AGCCTCTTCGCTTTTACCCC	

**Table 4 genes-16-00921-t004:** Primers used for mutagenesis.

Plasmid	Primers (5′-3′)	Mutagenesis
Circ_0001591_1	F: CTTAAGCCTCTGGG CAGCCCAGTAGGCC	Δ GAAT
	R: GGCCTACTGGGCTGCCCAGAGGCTTAAG	
Circ_0001591_2	F: TGGAATACTTTGAAAATACTTCAAGGCTGTAAGGTACTCATTAAAATAAAG	Δ CACT
	R: CTTTATTTTAATGAGTACCTTACAGCCTTGAAGTATTTTCAAAGTATTCCAC	
Circ_0001591_3	F: GTCCCCTCCCCCCAGAGGGACTTC	Δ AATG
	R: GAAGTCCCTCTGGGGGGAGGGGAC	
Circ_0001591_4	F: AGCTGACACAGTGCGCACTCCAGCCTTG	Δ CACT
	R: CAAGGCTGGAGTGCGCACTGTGTCAGCT	
AXL_miR-20a-3p	F: CCTTCAAGCCTGTGCAAT TAGGGATGCCTCCTTT	Δ GCAT
	R: AAAGGAGGCATCCCTAATTGCACAGGCTTGAAGG	
AXL_miR-34a-5p	F: GATCCAAGCTAAGCACTCTGGGGAAAACTCCACC	Δ GCCA
	R: GGTGGAGTTTTCCCCAGAGTGCTTAGCTTGGATC	
FOSL1_1	F: CCCCTTCCAGATCATATCTGCCACACTCTCC	Ins TAT
	R: GGAGAGTGTGGCAGATATGATCTGGAAGGGG	
FOSL1_2	F: ACTCACCAGCCCCACTGCGAGCAGCAGCAGGT	C→G
	R: ACCTGCTGCTGCTCGCAGTGGGGCTGGTGAGT	

## Data Availability

Data are contained within the article.

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
