# Peer review of "Circular RNA circ_0001591 Contributes to Melanoma Cell Migration Through AXL and FRA1 Proteins by Targeting miR-20a-3p and miR-34a-5p"

_genes, 2025, doi:10.3390/genes16080921_

Round 1
Reviewer 1 Report
Comments and Suggestions for Authors
After reading the manuscript my major concerns are as follows:
- Title of the article must be more informative. Please, avoid acronyms. Is it possible to replace “Circ_0001591” with a more informative fragment? Perhaps, Circular RNA …….?
- Please, consider to add a new Figure illustrating possible various molecular mechanisms linked to Axl and Fra1 signaling in melanoma. It would be more informative for potential readers of this article.
- Please, state clearly or confirm that all three melanoma cell lines used (A375, LM-36 and LM-20) have a BRAF mutation.
- Figure 1B, 1C, 1D, 1E, 1F and 1G. Please, recalculate the data presented in this Figure due to improper statistical test used. Statistical analysis of data should be performed with one-way ANOVA with repeated measure on time. The Student’s t-test is inappropriate because the same variable was estimated several times: at 4, 8, 12, 16 and 24 hrs (Figure 1B-1D) or 24, 48 and 72 hrs (Figure 1E-1G).
- The same holds true for multipart Figure 2, 4, 5. One-way ANOVA with repeated measure on time. Please, correct statistical analysis of data. Please, add F statistics with the respective degree of freedoms.
Author Response
Thank you very much for taking the time to provide in-depth and constructive feedback on our manuscript. We appreciate your feedback, which has helped us improve the quality of our submission. Below, you will find our answers to all your observations. All revisions and additions made to the manuscript have been highlighted.
Comment 1: Title of the article must be more informative. Please, avoid acronyms. Is it possible to replace “Circ_0001591” with a more informative fragment? Perhaps, Circular RNA …….?
Response 1: We agree that the title could be more informative. We have revised it, accordingly avoiding acronyms and replacing to “Circular RNA has_circ_0001591 contributes to melanoma cell migration through AXL and FRA1 proteins, by targeting miR-20a-3p and miR-34a-5p”.
Comment 2: Please, consider to add a new Figure illustrating possible various molecular mechanisms linked to Axl and Fra1 signaling in melanoma. It would be more informative for potential readers of this article.
Response 2: Thank you for this comment. We add a figure in the introduction (Figure 1), after the paragraph in which we explain AXL and FRA1 (line 78, page 2).
Comment 3: Please, state clearly or confirm that all three melanoma cell lines used (A375, LM-36 and LM-20) have a BRAF mutation.
Response 3: Thank you for pointing this out. We have added the sentence “All cell lines used carry the V600E mutation in the BRAF gene” in the Cell Culture paragraph (lines 377–378, page 14).
Comment 4: Figure 1B, 1C, 1D, 1E, 1F and 1G. Please, recalculate the data presented in this Figure due to improper statistical test used. Statistical analysis of data should be performed with one-way ANOVA with repeated measure on time. The Student’s t-test is inappropriate because the same variable was estimated several times: at 4, 8, 12, 16 and 24 hrs (Figure 1B-1D) or 24, 48 and 72 hrs (Figure 1E-1G).
Response 4: We thank the reviewer for the thoughtful suggestion regarding the statistical analysis in Figures 1B–1G (in the new version Figure 2).
In these experiments, we monitored scratch-wound closure over time, comparing a negative control to a single treatment group (siRNA transfection). Although multiple time points were recorded from the same experiment, our analysis focused on evaluating the effect of the treatment at each specific time point, rather than modeling the overall time-dependent response. For this reason, we performed pairwise comparisons at each time point using paired t-tests, as the same cell monolayers were followed over time.
We fully acknowledge the reviewer’s point: a repeated-measures ANOVA would be more appropriate for evaluating the interaction between treatment and time in a longitudinal design. However, that was not the primary goal of these analyses.
Comment 5: The same holds true for multipart Figure 2, 4, 5. One-way ANOVA with repeated measure on time. Please, correct statistical analysis of data. Please, add F statistics with the respective degree of freedoms.
Response 5: We thank the reviewer for the thoughtful observation regarding the statistical analysis of Figures 2, 4, and 5 (in the new version Figure 3, 5, and 6). As also explained in our response to Comment 4, our statistical choices were guided by the structure and purpose of each experiment.
Figure 2 (in the new version Figure 3) presents qRT-PCR results showing the expression of miR-20a-3p and miR-34a-5p following si-circ_0001591 transfection in three different melanoma cell lines (A375, LM-20, and LM-36). The data derive from independent biological replicates, each performed in separate wells, and include internal normalization for each cell line.
Similarly, Figure 4 (in the new version Figure 5) reports the expression of the same microRNAs following mimic or inhibitor transfection across the same three melanoma cell lines. Also in this case, measurements were taken from biologically independent cultures, each treated and analyzed as a distinct experimental unit.
Because no repeated measurements were taken on the same sample over time, we considered that repeated-measures ANOVA was not applicable for Figures 2 and 4 (in the new version Figure 3 and 5) . Instead, we focused on comparisons between conditions within each cell line at a single time point.
In Figure 5 (in the new version Figure 6), the design involved imaging the same wells over multiple time points after treatment, similar to the approach in Figure 1 (in the new version Figure 2).. Therefore, the same considerations regarding the possibility of applying a repeated-measures ANOVA (as discussed in Comment 4) would apply here as well.
Reviewer 2 Report
Comments and Suggestions for Authors
In this study, the Authors performed functional analyses that showed that circ_0001591 can act as a competitive endogenous RNA by sponging miR-20a-3p and miR-34a-5p. These microRNAs, in turn, regulate the expression of Axl and Fra1 oncoproteins by directly targeting the 3’UTR of AXL and FOSL1 mRNAs. In conclusion, our findings demonstrate that the circ_0001591, sponging miR-20a-3p and 23 miR-34a-5p, can indirectly modulate the expression of Axl and Fra1 oncoproteins, promoting melanoma migration.
Specific comments:
- Please use consistent and recommended style of gene and protein names as nicely explained e.g., https://www.jci.org/kiosk/publish/genestyle.
- Were cell lines authenticated and tested against Mycoplasma contamination?
- Wording should be revised, e.g., line 465-466 'circ_0001591 plays a role in melanoma cell line migration' should be rephrased 'circ_0001591 plays a role in melanoma cell migration.' as migration is a cellular feature, not assigned for a cell line as such.
Author Response
Thank you very much for taking the time to provide in-depth and constructive feedback on our manuscript. We appreciate your feedback, which has helped us improve the quality of our submission. Below, you will find our answers to all your observations. All revisions and additions made to the manuscript have been highlighted.
Comment 1: Please use consistent and recommended style of gene and protein names as nicely explained e.g., https://www.jci.org/kiosk/publish/genestyle.
Response 1: Thank you for this pertinent observation. We have systematically revised the nomenclature and formatting of all gene and protein names throughout the manuscript to ensure. All these modifications have been highlighted in yellow within the manuscript file for ease of identification and review.
Comment 2: Were cell lines authenticated and tested against Mycoplasma contamination?
Response 2: This is an important point, and we appreciate you raising it. Yes, all cell lines utilized in this study were tested for mycoplasma contamination. A clarifying sentence to this effect has been inserted into the "Cell Cultures" paragraph, specifically at lines 382-383, page 14.
Comment 3: Wording should be revised, e.g., line 465-466 'circ_0001591 plays a role in melanoma cell line migration' should be rephrased 'circ_0001591 plays a role in melanoma cell migration.' as migration is a cellular feature, not assigned for a cell line as such.
Response 3: Thank you for your comment. We changed the sentence as you suggested in line 498-499, page 17 (new version).